# Posterior and Para-Aortic (D2plus) Lymphadenectomy after Neoadjuvant/Conversion Therapy for Locally Advanced/Oligometastatic Gastric Cancer

**DOI:** 10.3390/cancers16071376

**Published:** 2024-03-31

**Authors:** Daniele Marrelli, Stefania Angela Piccioni, Ludovico Carbone, Roberto Petrioli, Maurizio Costantini, Valeria Malagnino, Giulio Bagnacci, Gabriele Rizzoli, Natale Calomino, Riccardo Piagnerelli, Maria Antonietta Mazzei, Franco Roviello

**Affiliations:** 1Unit of General Surgery and Surgical Oncology, Department of Medicine, Surgery and Neurosciences, University of Siena, 53100 Siena, Italy; stefipiccioni@gmail.com (S.A.P.); ludovicocarbone1@gmail.com (L.C.); gabriele.rizzoli.gr@gmail.com (G.R.); r.piagnerelli@ao-siena.toscana.it (R.P.); franco.roviello@unisi.it (F.R.); 2Unit of Medical Oncology, Department of Medicine, Surgery and Neurosciences, University of Siena, 53100 Siena, Italy; r.petrioli@ao-siena.toscana.it; 3Pathology Unit, University Hospital of Siena, 53100 Siena, Italy; costantini@ao-siena.toscana.it (M.C.); valeria.malagnino@ao-siena.toscana.it (V.M.); 4Unit of Diagnostic Imaging, Department of Medicine, Surgery and Neurosciences, University of Siena, 53100 Siena, Italy; giuliobagnacci@gmail.com (G.B.); mariaantonietta.mazzei@unisi.it (M.A.M.); 5Unit of Kidney Transplantation, Department of Medicine, Surgery and Neurosciences, University of Siena, 53100 Siena, Italy; natale.calomino@unisi.it

**Keywords:** D2plus lymphadenectomy, gastric cancer, neoadjuvant, conversion, posterior nodes, para-aortic nodes, survival

## Abstract

**Simple Summary:**

Surgery with adequate lymphadenectomy (D2) currently represents the standard of care for resectable gastric cancer under most guidelines. However, super-extended lymphadenectomy (D2plus) may offer better locoregional control in advanced stages with a high risk of metastases to third-level nodes. In recent years, preoperative chemotherapy has become a novel issue in patients with locally advanced gastric cancer. To date, only a few studies have evaluated D2plus lymphadenectomy in patients with locally advanced or oligometastatic gastric cancer after preoperative therapy. The present study included a large series when compared with the current literature and reports limited morbidity/mortality rates and relevant survival outcomes.

**Abstract:**

Super-extended (D2plus) lymphadenectomy after chemotherapy has been reported in only a few studies. This retrospective study evaluates survival outcomes in a Western cohort of locally advanced or oligometastatic gastric cancer patients who underwent D2plus lymphadenectomy after neoadjuvant chemotherapy. A total of 97 patients treated between 2010 and 2022 were included. Of these, 62 had clinical stage II/III disease, and 35 had stage IV disease. Most patients (65%) received preoperative DOC/FLOT chemotherapy. The mean number of lymph nodes harvested was 39. Pathological positive nodes in the posterior/para-aortic stations occurred in 17 (17.5%) patients. Lymphovascular invasion, ypN stage, clinical stage, and perineural invasion were predictive factors for positive posterior/para-aortic nodes. Postoperative complications occurred in 21 patients, whereas severe complications (grade III or more) occurred in 9 cases (9.3%). Mortality rate was 1%. Median overall survival (OS) was 59 months (95% CI: 13–106), with a five-year survival rate of 49 ± 6%; the five-year OS after R0 surgery was 60 ± 7%. In patients with positive posterior/para-aortic nodes, the median OS was 15 months (95% CI: 13–18). D2plus lymphadenectomy after chemotherapy for locally advanced or oligometastatic gastric cancer is feasible and associated with low morbidity/mortality rates. The incidence of pathological metastases in posterior/para-aortic nodes is not negligible even after systemic chemotherapy, with poor long-term survival.

## 1. Introduction

Gastric cancer (GC) remains a leading cause of neoplasm death worldwide, and lymph node spread is a prominent prognostic factor. Currently, D2 lymphadenectomy is considered the standard approach to resectable forms [1,2,3]. Although more extensive lymph node dissection (D2plus) has not shown a significant survival benefit when performed in a prophylactic setting [4], non-negligible long-term results have been observed in small subgroups of patients with pathological metastases to posterior (8p, 12b/p, and 13) and para-aortic nodal stations (16 a2/b1), which are classified as distant metastases due their extra-regional locations [5,6,7,8].

The prognosis of patients with GC is stage-dependent. Locally advanced stages (pT3-T4a, pN2-N3, and M0) have shown poor long-term survival despite R0 resection and well-performed lymphadenectomy [9,10,11]. In recent years, neoadjuvant chemotherapy (NAC) has been introduced to improve the results of upfront surgery, and it has been included in most Western guidelines as the standard of care for locally advanced GC [1,3]. Furthermore, conversion therapy has recently been proposed as an emerging option for metastatic forms, with complete or partial responses to medical treatment [12,13,14,15]. In these cases, D2 lymphadenectomy is generally recommended in radical surgery after chemotherapy. However, the removal of posterior and/or para-aortic clinically involved lymph nodes after preoperative chemotherapy has been correlated with improved survival outcomes in recent reports [16,17,18,19,20]. Furthermore, the recent Japanese Gastric Cancer Association (JGCA) guidelines suggest that we “consider” para-aortic lymphadenectomy in patients with bulky “regional” lymph nodes treated with NAC [2]. Currently, most studies evaluating the impact of D2plus lymphadenectomy after systemic chemotherapy are from East Asia. These include some phase II studies with limited series [17,21].

The aim of the present study is to analyze the results of D2plus lymphadenectomy after NAC or conversion therapy in a specialized Western center.

## 2. Materials and Methods

### 2.1. Patients

The present study included patients with locally advanced or oligometastatic GC who underwent preoperative chemotherapy followed by gastrectomy and D2plus lymphadenectomy between 2010 and 2022 at the Unit of General Surgery and Surgical Oncology (University of Siena, Italy). Inclusion criteria were as follows: histologically confirmed primary GC, locally advanced or oligometastatic clinical stages, at least two cycles of preoperative chemotherapy performed, and gastrectomy with lymphadenectomy beyond the D2. Exclusion criteria were the following: evidence of multiple-site metastasis on preoperative work-up, poor general conditions, palliative surgery, D2 or less extended lymphadenectomy, and Siewert type I/II carcinoma. A total of 97 patients were included: 64 males, median age 65 years (interquartile range (IQR): 57–70).

### 2.2. Preoperative Work-up and Chemotherapy

Preoperative work-up included upper-digestive endoscopy and biopsy, a computed tomography (CT) scan of the chest and the abdomen, endoscopic ultrasound (optional), and staging laparoscopy in cases of suspected peritoneal dissemination. Following clinical staging, all cases were discussed by the Institutional Upper-GI Multidisciplinary Team to determine the optimal therapeutic course. Indications for NAC or conversion therapy were based on the clinical stage, the patient’s general condition, and the absence of tumor complications (stenosis; bleeding). In general, chemotherapy was started within 2 weeks of the Multidisciplinary Team’s meeting. The decision regarding the chemotherapy schedule and number of cycles was based on the patient’s condition, clinical stage, and response to chemotherapy. Radiological restaging (CT scan) was performed after 3–4 cycles according to different schedules. At the end of chemotherapy, each case was re-evaluated by the Institutional Multidisciplinary Team.

### 2.3. Surgical Treatment

The interval between the end of chemotherapy and surgery was generally 4 weeks. The goal of surgery was complete resection (R0) of the tumor. Complete exploration of the peritoneal cavity and lavage of the peritoneum were always performed. The decision regarding resectability was based on the possibility of an R0 resection or a clinically evident response to chemotherapy in metastatic cases (conversion surgery). Distal subtotal gastrectomy was performed in the lower/middle third of the tumor, with a proximal resection margin of at least 5 cm from the tumor. The surgical technique of lymphadenectomy has been previously detailed [22]. The procedure included the systematic removal of the perigastric lymph node stations (nos. 1 to 7) and second-level nodes: celiac axis (no. 9), hepatic artery (no. 8a), splenic artery (no. 11p/d), and hepatoduodenal ligament (no. 12a). Splenectomy was only performed in cases with macroscopic involvement at the splenic hilum. All patients in this study also underwent systematic removal of the “posterior” lymph node stations located behind the hepatic artery (no. 8p), the hepatoduodenal ligament (no. 12b/p), the retropancreatic nodes (no. 13), and the para-aortic area (no. 16a2/b1).

### 2.4. Lymph Node Mapping and Pathological Classifications

Single lymph nodes were retrieved from the fresh specimen by the surgeon after the surgical procedure, divided into nodal stations according to the Japanese Gastric Cancer Association (JGCA) pathological classification, and sent for pathological examination. A dedicated pathologist performed the analysis. Each node was sectioned at the largest size plane, embedded in paraffin, cut into two planes with two sections per level, and stained with hematoxylin and eosin. The total number of dissected and metastatic nodes was recorded for each station. Histological type was defined according to Lauren and reclassified according to the latest WHO classification [23]. Lymphovascular and perineural invasions were also recorded. Staging for depth of invasion and nodal status was performed according to the 8th TNM. Tumor regression was classified according to the Becker criteria. Microsatellite instability and Her-2 status were also analyzed using standard techniques, as previously described [24].

### 2.5. Additional Treatments and Follow-Up

Additional postoperative treatment was decided by the tumor board based on the ypTNM stage and the patient’s general condition and response to treatment. After completion of treatment, patients were enrolled in a follow-up protocol.

Survival data were available for all patients. Follow-up was completed in September 2023. The median (IQR) follow-up period was 26 months (14–59) for the whole series and 50 months (20–92) for patients alive at the last follow-up.

### 2.6. Statistical Analysis

Statistical analysis was performed using the SPSS 26.0 statistical software (IBM Corp., Chicago, IL, USA). Correlations between the patients’ characteristics and positive “posterior” or para-aortic nodes were assessed with the χ^2^ test for categorical variables and the t-test or the non-parametric test (Mann–Whitney) for continuous variables. Survival probabilities were estimated using the Kaplan–Meier method considering the interval between the start of NAC and the date of death or the lost follow-up control for survivors. Differences between survival probabilities were compared using the log-rank test. The level of statistical significance was established at *p* < 0.05.

## 3. Results

### 3.1. Clinicopathological Characteristics and Preoperative Chemotherapy

The clinical and pathological features of the included patients are shown in Table 1. Although the median age was 65 years, 14 patients were older than 75 years. Most patients (68%) had neoplasms located in the lower two-thirds of the stomach; 62 cases were locally advanced forms and underwent NAC, whereas 35 cases were clinical stage IV and underwent conversion therapy prior to surgery. Triplets with taxanes (DOX-FLOT) were the most common schedule in our series. At clinical restaging, a response to chemotherapy was observed in 57.7% of patients, with only four cases of progressive disease.

### 3.2. Surgical Treatment and Lymphadenectomy

All patients in this series were treated with open surgery due to the retroperitoneal approach to D2plus lymphadenectomy. A total gastrectomy was performed in most cases (53.6%), and an extended resection to adjacent organs was performed in 12.4% of cases. At the end of the operation, an R0 resection was obtained in 68% of cases; in 20 patients, a microscopic residual tumor (R1) was found (positive peritoneal cytology; involvement of the resection margins), and 11 patients were R2 due to an incomplete response to chemotherapy in stage IV cases. 

After D2plus lymphadenectomy, a median (IQR) of 39 (30–56) total lymph nodes were removed; of these, a median of 25 (17–35) total lymph nodes were removed at the first level (stations 1 to 7), 6 (3–10) at the second level (stations 8 to 12), and 7 (4–13) at the third level (stations 8p, 12 b/p, 13, and 16) (Figure 1).

### 3.3. Metastases to Posterior and Para-Aortic Lymph Nodes

Overall, 32% of patients had negative lymph nodes after NAC, and 68% of cases were ypN-positive; the median (IQR) number of positive lymph nodes was two (0–8). Of the 97 patients who underwent D2plus lymphadenectomy, 17 (17.5%) had pathologically positive nodes in the third-level stations. Table 2 shows the correlation between clinicopathological factors and posterior/para-aortic lymph node metastases. Of all the variables considered, the total number of positive nodes (ypN stage), lymphovascular invasion, clinical stage, and perineural invasion were significant predictive factors for metastasis to these distant nodes. Notably, about 50% of cases with more than six positive nodes after NAC (ypN3 stage) had metastases in lymph nodes beyond the D2 area. However, even in the group with 3–6 positive nodes (ypN2), the incidence was about 19%; conversely, no distant nodal metastases were found in the ypN1 group (1–2 total nodes involved). 

Regarding the ypT stage, patients with complete tumor response or ypT1 tumors had no metastases in posterior/para-aortic stations, whereas the ypT2–T4 stages showed a similar incidence. Age, gender, and tumor location had no effect on distant lymph node metastases. Despite the higher incidence rates in the diffuse–mixed and poorly cohesive histotypes, even the intestinal and tubular/papillary types showed metastases in the posterior/para-aortic area. No significant difference was observed according to Her-2 status, while data regarding microsatellite instability were unreliable due to the small number of MSI tumors (five cases in our series).

### 3.4. Early Postoperative Outcomes

Postoperative complications occurred in 21 out of 97 included patients (morbidity rate: 21.6%) (Table 3); 10 patients had surgical complications (anastomotic leakage in 7), and medical complications were registered in 11 (above all, pleuro-pulmonary). Complications were graded as Clavien–Dindo III or higher in nine cases (9.3%), most of which resolved favorably with medical or conservative treatment. Reoperation was required in only two cases; one death was observed (due to cardiac arrest after surgery). The median postoperative hospital stay was 11 ± 3 days for the whole population; this value increased to 15 ± 12 days in complicated cases.

### 3.5. Long-Term Outcomes

At the end of follow-up, 43 patients died, 6 were alive with disease, and 48 were disease-free. The median overall survival (OS) for the entire population was 59 (95% CI: 13–106) months, with a five-year survival rate of 49 ± 6%. 

Figure 2a shows OS curves based on posterior/para-aortic lymph node involvement. As expected, a statistically significant difference was found between negative and positive cases (*p* < 0.001); the median OS of patients with pathologically positive distant nodes was 15 months (95% CI: 13–18). 

On the other hand, patients with positive posterior/para-aortic nodes showed similar survival rates compared to GC at the ypN3 stage (i.e., more than six total nodes involved) and worse 5-year OS than the ypN0–2 subgroups (Figure 2b).

In Table 4, the median and 5-year survival probability of the patients has been stratified according to clinicopathological variables. Surgical radicality was one of the most important prognostic factors: a 60% 5-year survival after R0 resection vs. 31% in R1 and no long-term survivors for R2. Other relevant factors were ypN (*p* < 0.001), tumor location (worse prognosis in the upper third; no survivors in the linitis plastica; *p* < 0.001), clinical stage (*p* = 0.006), and perineural invasion (*p* = 0.004). The overall impact of the ypT stage and the Becker regression grade was less than other factors; ypT4 and Becker grade 3 showed worse survival in these subgroups.

## 4. Discussion

D2 lymphadenectomy is the standard treatment in radical surgery after NAC or conversion surgery for locally advanced or oligometastatic GC. However, D2 lymphadenectomy alone may not be suitable in patients with bulky N2 and/or PAN metastases. There are several clinical considerations that may justify a lymphadenectomy beyond D2 to further improve prognosis. First, in patients with extra-regional lymph node metastases, especially in the para-aortic or retroperitoneal area (stations 12 b/p, 13, and 16), several studies have reported a chance of long-term survival after upfront surgery [5,6,8]. The Italian Research Group for Gastric Cancer (GIRCG) reported a 10.8% incidence of metastases to para-aortic nodes, with 11% five-year survival in positive cases in a large multicenter study [7]. As NAC has been reported to be associated with improved outcomes when compared with upfront surgery, it is expected that a multimodal approach could achieve more favorable results [25,26]. Indeed, several studies from East Asia have reported five-year survival rates exceeding 50% in patients with clinical para-aortic metastases who underwent NAC and down-staged after medical treatment [17,20]. Second, although this phenomenon has not been fully clarified, the potential benefit of extended lymphadenectomy in GC has been reported not only in patients with lymph node metastases in the resected field but even in cases without such involvement or node-negative stages [27,28]. Indeed, the JGCA guidelines recommend considering NAC and para-aortic lymphadenectomy in patients with bulky regional node metastases [2]. Third, the systemic disease control provided by NAC (reduction in early recurrence) [25], combined with the local control provided by more extended lymphadenectomy (reduction in late recurrence), may result in improved OS in patients treated with a multimodal approach [29,30]. 

The present study focused on the results of super-extended lymphadenectomy after NAC or conversion therapy in a Western referral center. As far as we know, only a few previous studies have investigated D2plus lymphadenectomy after NAC or conversion therapy by enrolling a wide number of patients. In general, preoperative chemotherapy has been reported to be associated with an increased risk of postoperative medical or surgical complications [31,32]. Edema and fibrosis in the retroperitoneal area induced by NAC can increase the technical difficulty of vascular skeletonization, as reported in our previous study [33]. The results of this study demonstrated the feasibility of this procedure even after NAC or extensive systemic chemotherapy (conversion). The morbidity rate was acceptable, with overlapping rates when compared with upfront surgery [7]. Most complications were treated with non-surgical management, with only two cases requiring reoperation. One death was observed due to cardiac arrest on the first postoperative day. Other experiences, mainly from Japan, have reported similar results. In particular, the JCOG 1704 trial enrolled 47 patients who received three cycles of docetaxel, oxaliplatin, and S-1 followed by D2plus lymphadenectomy: postoperative morbidity for grade III or higher was 11%, with no postoperative deaths [21].

Despite potential technical difficulties, the median number of lymph nodes removed with D2plus lymphadenectomy was 39. This number is not significantly different from that obtained with upfront surgery in the previous large multicenter GIRCG trial (median: 41 nodes dissected) [7]. However, the median number of positive nodes was only two, which is consistent with the nodal downstaging achieved with preoperative chemotherapy, as previously observed in the Italian GASTRODOC trial [34]. Of the 97 patients who underwent D2plus lymphadenectomy, 17 (17.5%) had pathologically positive nodes in the third-level stations, which is a high rate even when compared with upfront surgery. Most of the positive cases had a total of six or more involved lymph nodes (ypN3), lymphovascular or perineural invasion, and were stage IV on pre-treatment clinical staging. It is also interesting to note that histotype did not significantly affect metastases to lymph nodes beyond D2, with poorly cohesive (signet or non-signet ring cell subtypes) and tubular types being similarly involved.

The long-term outcome for patients from this series, including regarding locally advanced or oligometastatic GC, was also relevant: median survival, 59 months; 5-year survival, 49%. Although the pre-treatment clinical stage is not fully reliable, this rate appears to be significantly higher compared with upfront surgery at similar pTNM stages [9]. Our OS curves are better than those in the MAGIC trial and more similar to the FLOT trial, both of which included locally advanced forms without distant metastases [25,26]. However, the long-term outcome for patients with pathologically positive third-level nodes was dismal (median survival: 15 months; five-year survival: 8%), although this was not significantly different from the ypN3 stages. This is consistent with recent results from Japanese studies reporting low survival in patients with pathologically persistent para-aortic lymph node metastases after systemic chemotherapy [17,18,20].

In this study, 35 patients with clinical stage IV GC were treated with systemic chemotherapy followed by surgery, with a median survival of 19 months and a 5-year survival rate of 23%. This result confirms that stage IV patients with a good response to systemic chemotherapy could be offered a significant chance of cure, as previously observed in multicenter trials [15,35].

A peculiarity of the present report is that, at our center, the use of triplets with taxanes for NAC started in 2010, at the time of the GASTRODOC randomized trial; therefore, we were able to analyze the results of DOC/FLOT schedules with a long follow-up [34]. As reported in the FLOT trial, triplets with taxanes are associated with better outcomes than other schedules [26]. This was confirmed in the present study (54% five-year survival with DOC-FLOT vs. 32% with ECF-EOX); however, the number of patients treated with schedules other than DOC-FLOT was too small for a reliable comparison. 

The best survival results in the present study were observed in radically resected patients (R0), but even in the groups with microscopic residual tumors (R1), a chance of cure can be achieved (median survival: 23 months; 5-year survival 31%). In contrast, all R2 patients died after surgery, with a median survival of 14 months. This confirms the poor outcome after surgery with macroscopic tumor residuals in stage IV GC where conversion therapy has been attempted [35]. No significant impact of histotype on long-term outcomes was observed, although a poorly cohesive non-signet ring cell type was associated with worse survival. The predominant use of triplets with taxanes and the potential impact of D2plus lymphadenectomy on poorly cohesive–diffuse histotypes may help to explain such findings [7,26,36].

Perineural invasion was found to be a strong risk factor for metastasis to posterior/para-aortic nodes and a relevant predictor of poor long-term survival. It is considered an emerging prognostic factor in GC, and this study confirms the aggressive behavior of cases with perineural invasion, even in patients undergoing NAC or conversion therapy [37].

This study has several limitations. First, it is a retrospective study, although the clinical, pathological, and follow-up data were collected prospectively, and lymph node mapping was performed on fresh specimens, as has been the case for many years in our study group. Secondly, the lack of a control group limits the analysis of the real clinical impact of the procedure performed. The selection of patients who could benefit from a more extended approach is an important aspect that needs to be confirmed in future studies, as does the impact of the molecular type on the response to preoperative chemotherapy and related outcomes [38,39].

## 5. Conclusions

In conclusion, high survival rates can be achieved in locally advanced or oligometastatic GC treated with NAC/conversion therapy and D2plus lymphadenectomy. Patients with clinically positive PAN metastases should undergo preoperative chemotherapy followed by therapeutic PAN dissection. In the present study, performed at a specialized center, this procedure was feasible and resulted in a low morbidity risk. However, the additional benefit of D2plus lymphadenectomy should be investigated in multicenter comparative studies with D2 alone. Surgeons should weigh the potential oncological value of this extended procedure with the risk of postoperative complications and mortality. 

## Figures and Tables

**Figure 1 cancers-16-01376-f001:**
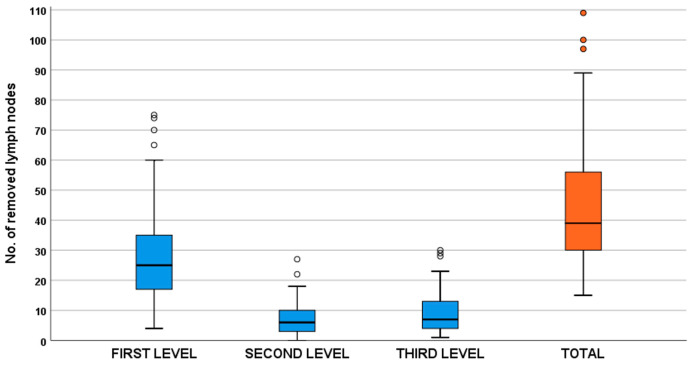
Number of lymph nodes removed in the study population divided by first-, second-, and third-level stations.

**Figure 2 cancers-16-01376-f002:**
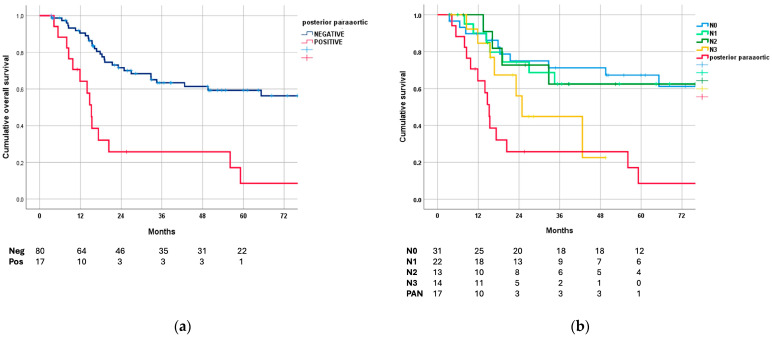
Survival curves: (**a**) overall survival based on posterior/para-aortic lymph node involvement; (**b**) overall survival based on ypN stage compared with positive posterior/para-aortic nodes at pathological evaluation.

**Table 1 cancers-16-01376-t001:** Clinical, pathological, and therapeutic features of the study population.

Characteristics	No. (%)
**Age** [IQR]	65 (26–83)
**Gender**	
Male	64 (66)
Female	33 (34)
**Location**	
Upper	23 (23.7)
Middle	23 (23.7)
Lower	43 (44.3)
Linitis	8 (8.3)
**Clinical Stage**	
II	14 (14.4)
III	48 (49.5)
IV	35 (36.1)
**Lauren Classification**	
Intestinal	45 (46.4)
Diffuse/mixed	47 (48.5)
NC	5 (5.1)
**WHO Classification**	
Mucinous	5 (5.2)
Tubular/papillary	40 (41.2)
Poorly cohesive-non SRC	13 (13.4)
Poorly cohesive-SRC	34 (35.1)
NC	5 (5.1)
**Her2 Status**	
Neg	87 (89.7)
Pos	9 (9.3)
NA	1 (1.0)
**Microsatellite Status**	
MSS	90 (92.8)
MSI	5 (5.1)
NA	2 (2.1)
**Chemotherapy**	
DOX/FLOT	63 (64.9)
ECF/EOX	19 (19.6)
FOLFOX	6 (6.2)
Others	9 (9.3)
**No. Chemotherapy cycles**	
2–4	77 (79.4)
5–8	14 (14.4)
>8	6 (6.2)
**Clinical Restaging**	
CR	4 (4.1)
PR	52 (53.6)
SD	37 (38.2)
PD	4 (4.1)
**Gastrectomy**	
Total	52 (53.6)
Subtotal	45 (46.4)
**Extended organ Resection**	
Performed	12 (12.4)
Not performed	85 (86.6)
**UICC R**	
R0	66 (68)
R1	20 (20.6)
R2	11 (11.4)
**Lymphovascular Invasion**	
Present	56 (57.7)
Absent	41 (42.3)
**Perineural Invasion**	
Present	49 (50.5)
Absent	48 (49.5)
**Pathological Stage**	
0	7 (7.2)
I	12 (12.4)
II	25 (25.8)
III	23 (23.7)
IV	30 (30.9)
**ypT**	
0	9 (9.3)
1	4 (4.1)
2	13 (13.4)
3	22 (22.7)
4	49 (50.5)
**ypN**	
0	31 (32.0)
1	22 (22.7)
2	16 (16.5)
3a	15 (15.4)
3b	13 (13.4)
**yM**	
0	67 (69.2)
M1 (cy+, peritoneal)	16 (16.4)
M1 (hematogenous)	5 (5.1)
M1 (extra-regional nodes)	9 ^1^ (9.3)
**Becker Regression Grade**	
1	17 (17.5)
2	23 (23.8)
3	49 (50.5)
NA	8 (8.2)

^1^ In 8 cases, lymph nodes were associated with other metastatic sites. IQR: interquartile range; NC: not classified; NA: not assessable; WHO: World Health Organization; SRC: signet ring cell; MSS: microsatellite stability; MSI: microsatellite instability; CR: complete response; PR: partial response; SD: stable disease; PD: progressive disease; cy+: positive cytology.

**Table 2 cancers-16-01376-t002:** Clinical–pathological risk factors for lymph node metastasis in posterior or para-aortic stations.

Characteristics	Posterior/Para-Aortic Lymph Node Metastasis Negative (No. = 80) No. (%)	Posterior/Para-Aortic Lymph Node Metastasis Positive (No. = 17) No. (%)	*p*
**Age**			0.965
<55	17 (85)	3 (15)	
55–64	22 (81.5)	5 (18.5)	
65–74	30 (83.3)	6 (16.7)	
75–84	11 (78.6)	3 (21.4)	
**Gender**			0.903
Male	53 (82.8)	11 (17.2)	
Female	27 (81.8)	6 (18.2)	
**Location**			0.882
Upper	19 (82.6)	4 (17.4)	
Middle	20 (87)	3 (13)	
Lower	35 (81.4)	8 (18.6)	
Linitis	6 (75)	2 (25)	
**Clinical Stage**			<0.001
II	14 (100)	0 (0)	
III	46 (95.8)	2 (4.2)	
IV	20 (57.1)	15 (42.9)	
**Lauren Classification**			0.255
Intestinal	39 (86.7)	6 (13.3)	
Diffuse/mixed	36 (76.6)	11 (23.4)	
NC	5 (100)	0 (0)	
**WHO Classification**			0.489
Mucinous	5 (100)	0 (0)	
Tubular/papillary	34 (85)	6 (15)	
Poorly cohesive-non SRC	10 (76.9)	3 (23.1)	
Poorly cohesive-SRC	26 (76.5)	8 (23.5)	
NC	5 (100)	0 (0)	
**Her-2 Status**			0.873
Neg	72 (82.8)	15 (17.2)	
Pos	7 (77.8)	2 (22.2)	
NA	1 (100)	0 (0)	
**Microsatellite Status**			0.178
MSS	76 (84.4)	14 (15.6)	
MSI	3 (60)	2 (40)	
NA	1 (50)	1 (50)	
**Lymphovascular Invasion**			0.005
Present	41 (73.2)	15 (26.8)	
Absent	39 (92.1)	2 (4.9)	
**Perineural Invasion**			0.018
Present	36 (73.5)	13 (26.5)	
Absent	44 (91.7)	4 (8.3)	
**ypT**			0.322
0	9 (100)	0 (0)	
1	4 (100)	0 (0)	
2	11 (84.6)	2 (15.4)	
3	19 (86.4)	3 (13.6)	
4	37 (75.5)	12 (24.5)	
**ypN**			<0.001
0	31 (100)	0 (0)	
1	22 (100)	0 (0)	
2	13 (81.25)	3 (18.75)	
3a	8 (53.3)	7 (46.7)	
3b	6 (46.2)	7 (53.8)	
**Becker**			0.236
1	16 (94.1)	1 (5.9)	
2	19 (82.6)	4 (17.4)	
3	37 (75.5)	12 (24.5)	
NA	8 (100)	0 (0)	

NC: not classified; NA: not assessable; WHO: World Health Organization; SRC: signet ring cell; MSS: microsatellite stability; MSI: microsatellite instability; Neg: negative; Pos: positive.

**Table 3 cancers-16-01376-t003:** Postoperative complications and related treatments in patients with p.o. morbidity.

Complications	No. (%)	Grade II ^1^	Grades III–IV ^1^	Treatment
SURGICAL	10 (10.3)	6	4	
Anastomotic leakage	7 (7.2)	3	4	Medical (4), surgery (2), other intervention (1)
Abdominal abscess	2 (2.1)	2	0	Conservative (2)
Lymphocele	1 (1.0)	1	0	Conservative (2)
MEDICAL	11 (11.3)	3	5	
Pleuro-pulmonary	7 (7.2)	3	4	Pharmacological (7)
Cardiovascular	2 (2.1)	1	1 **	Pharmacological (1), death (1)
Acute renal failure	2 (2.1)	2	0	Pharmacological (2)
90-day morbidity	21 (21.6)	12	9	
30-day mortality	1 (1.0)			
Hospital stay (days) ^2^	15 ± 2 *			

^1^ Clavien–Dindo classification; ^2^ mean ± standard deviation; * complicated cases (grades II-IV); ** grade V.

**Table 4 cancers-16-01376-t004:** Median survival and 5-year survival rates in relation to the clinical–pathological characteristics of the study population.

Characteristics	No.	Median Survival (Months, 95% CI)	5-Year Survival Rate (% ± SE)	*p*
**Age**				0.580
<65	50	34 (21–62)	46 ± 8	
>65	47	65 (26–104)	54 ± 9	
**Gender**				0.894
Male	33	56 (20–88)	47 ± 10	
Female	64	65 (9–120)	50 ± 7	
**Location**				<0.001
Upper	23	19 (16–22)	33 ± 10	
Middle	23	NR	58 ± 12	
Lower	43	NR	66 ± 9	
Linitis	8	15 (13–18)	0	
**Clinical Stage**				0.006
II	14	NR	70 ± 15	
III	48	NR	61 ± 8	
IV	35	19 (12–26)	23 ± 10	
**Lauren Classification** ^1^				0.273
Intestinal	45	85 (39–120)	55 ± 8	
Diffuse/Mixed	47	43 (10–75)	42 ± 9	
**WHO Classification** ^1^				0.738
Tubular/papillary	40	65 (23–108)	52 ± 9	
Poorly cohesive-non SRC	13	35 (0–69)	31 ± 16	
Poorly cohesive-SRC	34	43 (15–89)	47 ± 10	
**Chemotherapy Schedule**				0.557
DOX-FLOT	63	84 (40–110)	54 ± 7	
ECF-EOX	19	30 (10–43)	32 ± 11	
Others	15	NR	61 ± 15	
**Clinical Restaging**				0.106
CR+PR	56	85 (40–105)	56 ± 8	
SD+PD	41	25 (3–47)	40 ± 9	
**Gastrectomy**				0.030
Total	52	27 (11–43)	41 ± 7	
Subtotal	45	NR	60 ± 9	
**UICC R**				<0.001
0	69	NR	60 ± 7	
1	18	23 (6–40)	31 ± 13	
2	10	14 (8–20)	0	
**Lymphovascular Invasion**				0.040
Present	56	27 (5–49)	37 ± 8	
Absent	41	NR	63 ± 8	
**Perineural Invasion**				0.004
Present	49	21 (12–30)	31 ± 8	
Absent	48	NR	57 ± 9	
**ypT**				0.025
0	9	NR	67 ± 16	
1–2	17	NR	80 ± 13	
3	22	NR	52 ± 13	
4	49	25 (10–40)	34 ± 8	
**ypN**				<0.001
0	31	NR	67 ± 9	
1	22	85 (40–110)	62 ± 11	
2	16	NR	53 ± 14	
3	28	17 (11–22)	0	
**Becker** ^1^				0.306
1	17	NR	62 ± 15	
2	23	65 (10–120)	48 ± 12	
3	49	34 (3–68)	12 ± 8	

^1^ Patients who were not assessable (NV) or not classifiable (NC) were excluded. SE: standard error; NR: not reached; WHO: World Health Organization; SRC: signet ring cell; CR: complete response; PR: partial response; SD: stable disease; PD: progressive disease; UICC R: residual tumor.

## Data Availability

The data presented in this study are available on request from the corresponding author.

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
