# Peer review of "Posterior and Para-Aortic (D2plus) Lymphadenectomy after Neoadjuvant/Conversion Therapy for Locally Advanced/Oligometastatic Gastric Cancer"

_cancers, 2024, doi:10.3390/cancers16071376_

Round 1

Reviewer 1 Report

Comments and Suggestions for Authors

Dear Authors, excellent article I have no special comments! 

Author Response

Dear Reviewer, we appreciate your comment. Sincerely.

Reviewer 2 Report

Comments and Suggestions for Authors

D2 lymphadenectomy alone, currently, seems to be suitable for advanced Gastric Cancer (GC) patients with only bulky N2 metastases after preoperative chemotherapy (PCT). However, D2 lymphadenectomy alone perhaps is not suitable for patients with bulky N2 and/or para-aortic lymph node (PAN) metastases after PCT. In these patients it is unclear whether,on the basis of PCT , the addition of para-aortic lymphadenectomy (D2+) would further improve prognosis compared to D2 lymphadenectomy alone. Eastern and Western scholars have different opinions on the treatment of these patients. Nevertheless patients with clinically positive PAN metastases should undergo neoadjuvant chemotherapy followed by therapeutic PAN dissection. In western country debate exists in the cases of no evidence of positive PAN. Randomized controlled trials to explore the role of prophylactic D2+ compared to the standard D2 have been published. Nevertheless surgeons should balance possible oncological value of D2+ lymphadenectomy with the price of post-operative complications and the risk of mortality

Author Response

Dear Reviewer, thank you for your comments and suggestions. We agree with your statements, and modified "Discussion" and "Conclusions" as following:  

Discussion
D2 lymphadenectomy is the standard treatment in radical surgery after NAC or conversion surgery for locally advanced or oligometastatic GC. However, D2 lymphadenectomy alone may not be suitable in patients with bulky N2 and/or PAN metastases. There are several clinical considerations that may justify a lymphadenectomy beyond D2 to further improve prognosis.

Conclusion
In conclusion, high survival rates can be achieved in locally advanced or oligometastatic GC treated with NAC/conversion therapy and D2plus lymphadenectomy. Patients with clinically positive PAN metastases should undergo preoperative chemotherapy followed by therapeutic PAN dissection. In the present study, performed in specialized center, this procedure was feasible and resulted in a low morbidity risk. However, the additional benefit of D2plus lymphadenectomy should be investigated in multicentre comparative studies with D2 alone. Surgeons should weight the potential oncological value of this extended procedure with the risk of post-operative complications and mortality.  

Reviewer 3 Report

Comments and Suggestions for Authors

Very important data for me. Well written

Author Response

Dear Reviewer 3, we are very pleased to receive this comment. Sincerely.